# Evaluation of Temporomandibular Joint in Patients with Parkinson’s Disease: A Comparative Study

**DOI:** 10.3390/diagnostics13152482

**Published:** 2023-07-26

**Authors:** Ayse Selenge Akbulut, Aysun Hatice Akca Karpuzoglu

**Affiliations:** 1Department of Orthodontics, Peri Sokak Dental Clinic, 42010 Konya, Türkiye; 2Department of Neurology, Medova Hospital, 42070 Konya, Türkiye; aysun.karpuzoglu@medova.com.tr

**Keywords:** MR, orthodontics, Parkinson’s disease, TMD, TMJ

## Abstract

(1) The aim of this study was to perform an evaluation of the temporomandibular joint (TMJ) in patients with Parkinson’s disease (PD) and present the morphological differences of the TMJ between healthy subjects and patients with PD. (2) A total of 102 Caucasian subjects were divided equally into two groups. The study group consisted of patients with PD, while the control group comprised healthy subjects. Ten parameters, including anterior joint space (AJS), superior joint space (SJS), posterior joint space (PJS), condyle head length (CHL), condylar neck width (CNW), minor axis of the condyle (MAC), long axis of the condyle (LAC), condylar axis inclination (CI), medial joint space (MJS), and lateral joint space (LJS), were measured using magnetic resonance images. The data were statistically analyzed using paired samples *t*-test and Student’s *t*-test, with a significance level set at *p* < 0.05. (3) In the PD group, all TMJ parameters showed a statistically significant difference between both sides of the face (*p* < 0.05). However, in the control group, AJS, SJS, PJS, CHL, CNW, MAC, CI, MJS, and LJS did not show a statistically significant difference between both sides of the face (*p* > 0.05), except for LAC (*p* < 0.05). The asymmetry index values of AJS, SJS, PJS, CHL, CNW, MAC, CI, MJS, and LJS demonstrated a statistically significant difference between the study and control groups (*p* < 0.05), except for LAC (*p* > 0.05). (4) Within the limitations of this retrospective study, the findings suggest that TMJ morphology and asymmetry could be associated with PD.

## 1. Introduction

The temporomandibular joint (TMJ) is a specific, complex bilateral joint that connects the mandible to the temporal bone. Clinicians deal with various symptoms of temporomandibular disorders (TMDs), such as myofascial pain, TMJ sounds, restriction of mandibular movements, headache, or ear problems, in daily practice [1,2]. The prevalence of TMDs in adults and the elderly has been reported as 31.1% in a recent meta-analysis [3]. Considering this important prevalence rate of TMDs, the etiological factors underlying TMDs have also been gaining importance. Occlusal factors, psychological factors, hormonal factors, microtrauma, parafunctional habits, joint hyperlaxity and joint hypermobility, and hereditary factors are some of the etiological factors that take part in the occurrence of TMD [4].

Parkinson’s disease (PD) is a neurodegenerative disorder that affects approximately 6.1 million people worldwide [5], and its incidence is growing due to the increasing aging of the population. The exact cause of PD is not completely known, but it is recognized that a decline in dopamine levels occurs due to the degeneration of dopaminergic neurons in the substantia nigra [6]. While a cure for PD is not presently accessible, it is feasible to alleviate the symptoms by employing dopaminergic replacement therapy. Levodopa, which acts as a precursor to dopamine, is frequently employed in the medical management of PD symptoms [7]. Some specific gene mutations can be attributed to PD in approximately 5% to 10% of cases. However, the majority of individuals with PD do not possess these mutations [8,9]. This may indicate that PD may also be involved with undiscovered genetic mutations in addition to known genetic mutations, as well as other environmental and lifestyle-related factors, such as head trauma, exposure to chemical toxins, and smoking [9]. Environmental and genetic factors influence a shared network of pathways that involve oxidative stress, mitochondrial dysfunction, protein aggregation, neuroinflammation, and impaired autophagy [10,11].

Some of the motor and nonmotor symptoms of PD are bradykinesia, muscle stiffness, tremor, body imbalance, impaired posture and coordination, freezing of gate, constipation, fatigue, anxiety, cognitive impairment, depression, and sleeping disorders [12,13,14,15,16,17]. The presence of symptoms such as rigidity, muscle stiffness, and tremor in the masticatory muscles in PD can result in clenching, bruxism, and TMDs [18,19]. Additionally, given that PD predominantly affects the motor system, it is likely that the resulting motor symptoms directly contribute to the development of TMDs.

On the other hand, the complex neural network in the TMJ area prompts researchers to question its potential relationship with certain neurodegenerative diseases. Sensory signals from the TMJ are transmitted through the trigeminal nerve to the brainstem, where they are processed and relayed to higher brain regions responsible for pain perception and motor control. Motor neurons in the brainstem play a crucial role in the control of movement throughout the body, and their dysfunction can have significant implications in various neurological diseases [20,21]. From this perspective, instead of considering TMD solely as a symptom of PD, it is essential to investigate the TMJ to determine whether it can serve as an underlying biomechanical risk factor for PD.

Previous studies have focused on the relationship between TMJ and PD [22,23,24,25,26]. However, considering the important anatomical connections of TMJ, more studies that investigate the relationship between TMJ and PD on a morphological basis are needed for better understanding.

This study aimed to perform an evaluation of the TMJ in patients with PD and present the morphological differences of TMJ between healthy subjects and patients with PD. The null hypotheses were as follows:Patients with PD show symmetry in their bilateral temporomandibular joints;Healthy individuals show asymmetry in their bilateral temporomandibular joints;There is no difference in temporomandibular joint symmetry between healthy individuals and individuals with PD.

## 2. Materials and Methods

The protocol of this retrospective split-mouth study was approved by the ethical committee of Necmettin Erbakan University (2023/4325). Initial records of subjects referred to Medova Hospital, Department of Neurology (Konya, Turkey) between 2017 and 2023 were consecutively recruited.

Patients diagnosed with PD, patients between 40 and 95 years of age, and patients who had magnetic resonance (MR) images of the brain that also included the TMJ area clearly were selected for the study group. The health management information system enables users to filter patients based on different categories. In this case, the filter settings were adjusted to select only patients diagnosed with PD between the years 2017 and 2023. Additional filtering functions were then employed to meet the criteria of having brain MR images and specific gender requirements.

Healthy subjects between 40 and 95 years of age and subjects who had MR images of the brain that included the TMJ area clearly were selected for the control group. For selection of control group, the filter settings of health management information system were adjusted to select only check-up subjects with no specific disease between the years 2017 and 2023. From this group, subjects who had undergone brain MR scans were selected using the filter function. The subjects were then categorized based on their gender.

Subjects with craniofacial anomalies, a menton deviation greater than 2 mm, a history of traumatic injury, or a history of surgery in the craniofacial region were excluded from both the study and control groups. Menton deviation was measured in the coronal view. The perpendicular distance from the menton point to the midsagittal plane (MSP) was measured. If the distance was more than 2 mm, the subject was excluded from the study. History of traumatic injury and history of surgery in the craniofacial region were assessed through a nationwide patient record system, and the health management information system was provided by the hospital.

According to the power analysis conducted using G* Power software (version 3.1; Heinrich Heine University, Düsseldorf, Germany), it was determined that a minimum of 102 participants would need to be included in the study, considering a power of 0.80 and an effect size of 0.5. For the study group, 51 Caucasian patients were randomly selected, and for the control group, another 51 Caucasian subjects were also selected randomly using random allocation software.

The study was divided into three parts. In the first part, a morphological comparison of the TMJs between both sides was performed in the study group. In the second part, a morphological comparison of the TMJs between both sides was performed in the control group. The third part involved a morphological and asymmetrical comparison of the TMJs between the study and control groups.

### 2.1. Measurement Method

In the current study, MR images of the brain were utilized. The MR images were captured using the same digital machine (Siemens Magnetom Aera 1.5 Tesla, Siemens Healthcare, Erlangen, Germany) and operated by the same individual. T2-weighted images were acquired using a standardized method during the MR image acquisition process. Measurements were conducted using the same software (Ea Medware Pacs Version 3.0.0.0., EA software, Turkey) by a single investigator. To assess inter-examiner reliability, the measurements for five randomly selected subjects from study and control groups were repeated by another examiner.

The TMJ morphology was evaluated using sagittal, axial, and coronal views. The measurements were conducted following the modified version of measurements described in the literature [27,28,29]. The detailed description of the landmarks and parameters used in the current study is given in Table 1.

For the sagittal view parameters, the slices that exhibited the greatest anteroposterior extent of the condylar head were selected. The sagittal view parameters are as follows (Figure 1):Anterior joint space (AJS);Superior joint space (SJS);Posterior joint space(PJS);Condyle head length (CHL);Condylar neck width (CNW).

For axial view parameters, the slices that demonstrated the largest mesiodistal extent of the condylar head were selected in axial view. Axial view parameters are as follows (Figure 2):Minor axis of the condyle (MAC);Long axis of the condyle (LAC);Condylar axis inclination (CI).

For coronal view parameters, the slices that demonstrated the largest mesiodistal extent of each condylar head were selected in coronal view. The coronal view parameters and their descriptions are as follows (Figure 3):Medial joint space (MJS);Lateral joint space (LJS).

Additional asymmetry index parameters were calculated for each parameter based on Habets’ formula [30]. These parameters were utilized for the comparison between the study and control groups.

### 2.2. Statistical Analysis

The data were statistically analyzed using IBM SPSS Statistics Version 26.0 (Chicago, IL, USA). Pearson correlation analysis was performed to assess inter-examiner reliability. Confirmation of the data’s normal distribution was evaluated using the Kolmogorov–Smirnov test due to the sample size exceeding 30.

Before conducting the paired samples *t*-test, the necessary assumptions were checked. The data were collected in pairs, and each observation within a pair was dependent on the other. The data exhibited a normal distribution, and the differences between paired observations showed equal variances. The values within each pair were not influenced by or related to the values in other pairs. As these assumptions were met, the paired samples *t*-test was performed.

The assumptions underlying Student’s *t*-test were also checked. The normality assumption was met, and the variances were equal between the study and control groups. The observations within each group were independent of each other. Therefore, Student’s *t*-test was performed.

In the first part of the study, the comparison of TMJ parameters between both sides of the face was analyzed using a paired samples *t*-test in the study group. Similarly, in the second part, the comparison of TMJ parameters between both sides of the face was also analyzed using a paired samples *t*-test in the control group. The lowest values of each parameter were recorded under “Side 1”, while the highest values were recorded under “Side 2”. For the analyses in the first and second parts, the lowest and highest values were used instead of distinguishing between right and left sides. In the third part of the study, Student’s *t*-test was employed to compare the asymmetry indices of each TMJ parameter between the study group and the control group. A *p*-value of 0.05 was considered statistically significant.

## 3. Results

The correlation coefficients were 0.907 and 0.921 for the right side parameters and left side parameters, respectively (Table 2). These results indicated a high reliability between the measurements of the two examiners.

Furthermore, the overall data were found to be distributed normally based on the Kolmogorov–Smirnov test (*p* > 0.05). Consequently, parametric tests were applied in the present study. A paired samples *t*-test was conducted in the first and second parts of the study, while a Student’s *t*-test was performed in the third part of the current study.

Among the total number of subjects, 40.2% were females and 59.8% were males. In the study group, there were 21 females (41.2%) and 30 males (58.8%), while the control group consisted of 20 females (39.2%) and 31 males (60.8%). The demographic characteristics of the subjects are presented in Table 3.

Mean values of parameters for both TMJs in study and control groups are given in Table 4.

The comparison of TMJ parameters between both sides of the face through paired samples *t*-test is given in Table 5 for the study group. All parameters that were measured on the three planes of the face showed a statistically significant difference between both sides of the face in the PD group (*p* < 0.05).

The comparison of TMJ parameters between both sides of the face through paired samples *t*-test is given in Table 6 for the control group. AJS, SJS, PJS, CHL, CNW, MAC, CI, MJS, and LJS did not show a statistically significant difference between both sides of the face in control group. LAC showed a statistically significant difference between both sides of the face in control group (*p* < 0.05).

Comparison of asymmetry indices between study and control groups through student *t*-test is given in Table 7. Index values of AJS, SJS, PJS, CHL, CNW, MAC, CI, MJS, and LJS showed a statistically significant difference between study and control groups (*p* < 0.05). LAC index did not show a statistically significant difference between study and control groups.

## 4. Discussion

In the current study, a significant difference between the PD group and control group was observed with regard to TMJ morphology and symmetry.

The accurate imaging technique is crucial for assessment of the TMJ. Several imaging modalities have been utilized for evaluation of the TMJ area, including conventional radiography, computed tomography (CT), MR imaging, and cone-beam computed tomography (CBCT). Among these modalities, MR imaging offers exceptional quality images of soft tissues. It enables comprehensive evaluation of the soft tissue structures of the TMJ, synovial tissue, and articular disc. Therefore, the TMJ was evaluated through MR images in the present study.

In the first and second part of the current study, the comparisons were not performed based on the right and left sides of the face. The comparisons were performed based on the side that shows the highest values and the side that shows the lowest values of the parameters. Rather than focusing on right and left sides, the focus was on the difference between bilateral joints. In this way, the parameters that were effective in TMJ asymmetry were revealed through paired samples *t*-test. The statistical analyses that used in the current study allowed us to minimize personal differences between each subject. And, asymmetry indices for each parameter enabled us to compare study group and control group.

According to the results of the current study, healthy subjects showed morphological similarities and symmetry of the TMJ bilaterally in general (Figure 4). However, only the long axis of the condyle showed asymmetry in healthy subjects. On the other hand, patients with PD showed morphological differences and asymmetry of the TMJ bilaterally regarding all parameters measured in the three different planes (Figure 4). When the differences of both sides were compared between the healthy subjects and patients with PD, those with PD showed more asymmetrical TMJs in general except for the long axis of the condyle.

Although comparison of the parameters in PD group was impossible due to a lack of similar study design, the comparison of some parameters in control group was performed with previous studies that have control groups as well [27,29,31]. The mean value of AJS (3.4 ± 0.6) in the current study was higher compared to the previous studies (1.3 ± 0.2, 2.60 ± 0.79, 2.03 ± 0.50; and right: 2.1 ± 0.5; left: 1.9 ± 0.5) [27,31,32,33]. The higher values of AJS could be explained by the findings of a previous study that presents a tendency of increase in anterior space with age [34]. There was an important difference of sample age between the current study and above-mentioned previous studies. Therefore, lower values of AJS could be in accordance with the sample age between 12 and 59 years of age that was of previous studies’ age range. The mean value of SJS (3.5 ± 0.9) in the current study was similar compared to previous studies (3.35 ± 0.73, 4.13 ± 1.03; and right: 3.2 ± 0.9; left: 3.4 ± 0.9) [27,32,33]. The mean value of PJS (2.3 ± 0.9) in the current study was similar compared to previous studies (2.1 ± 0.3, 2.49 ± 0.73, 2.64 ± 0.81; and right: 2.1 ± 0.7; left: 2.4 ± 0.8) [27,31,32,33]. The mean value of CHL (10.0 ± 1.6) in the current study was similar compared to a previous study conducted on TMJ radiography (right: 10.83 ± 1.65; left: 11.53 ± 2.82) [35].

The mean value of MAC (8.0 ± 1.6) in the current study was similar compared to a previous study (8.36 ± 1.29) [31]. The mean value of CI (71.47 ± 7.14) in the current study was similar compared to previous studies (71.88 ± 7.38, 75.46 ± 4.58) [31,32]. However, the mean value of CI in the current study was higher than another previous study (66.46 ± 8.91) [36]. The inconsistency between two studies could be explained by the records, which were both MR and CT images in the previous study. Moreover, the subjects were between 11 and 44 years of age, which was a range younger than the subjects of the current study. Changes that occur by aging could also be a possible reason for this difference.

The mean value of MJS (3.0 ± 0.7) in the current study was similar compared to previous studies (2.54 ± 0.80, 2.75 ± 0.90; and right: 2.9 ± 1; left: 2.9 ± 0.9) [31,32,33]. The mean value of LJS (2.6 ± 0.7) in the current study was similar compared to a previous study (right: 2.6 ± 0.9; left: 2.4 ± 0.8) [33].

The association between PD and TMDs was assessed in previous studies with different methodologies. According to a previous survey study, 110 patients with PD were underwent comprehensive evaluation to show the frequency of TMD symptoms in patients with PD. It was revealed that the prevalence of TMDs was found to be 35%, exhibiting a higher occurrence among males (58%) and elderly individuals (53%) [24]. Higher occurrence of TMDs among patients diagnosed with PD was also presented in another previous cross-sectional study [23]. In a previous cohort study, it was reported that a 2.11-fold increase in TMD risk exists for individuals with PD compared to healthy people [25]. Another previous cohort study that involves a total of 514.866 participants consisted of two parts. According to Part I, a cohort of 4.455 individuals diagnosed with TMD was meticulously matched with 17.820 control participants, maintaining a ratio of 1:4. Similarly, in Part II, a group of 6.076 individuals with PD was carefully matched with 24.304 control participants, also at a ratio of 1:4 [37].

Although the aforementioned studies have revealed an association between PD and TMD, there is still a debate as to whether TMDs are a risk factor for PD, or whether PD is a risk factor for TMDs. In first scenario, considering the important neural network in the TMJ area, the improvement of PD symptoms after intervention in the TMJ may affirm that TMDs can be a risk factor for PD. Neural connections contribute to the sensory and motor functions of the TMJ, allowing for sensory perception, proper jaw movement, and control of the related muscles. The TMJ is innervated by branches of the trigeminal nerve (fifth cranial nerve), specifically its mandibular branch. This nerve provides sensory innervation to the TMJ, transmitting information related to pain, touch, and temperature. The sensory signals originating from the TMJ is transmitted to the trigeminal sensory nuclei situated in the brainstem. Within these nuclei, the sensory information is processed and then transmitted to higher brain regions that are responsible for tasks such as pain perception and motor control. The sensory input originating from the TMJ is further relayed to different regions of the central nervous system, including the thalamus and somatosensory cortex. This transmission enables the perception and interpretation of sensations arising from the TMJ, allowing for the brain to process and make sense of the sensory information received from the TMJ region. The motor control of the muscles related to the TMJ, such as the masticatory muscles, relies on the activation of motor neurons located in the brainstem. These motor neurons are responsible for transmitting signals to the muscles, thereby facilitating the movements of the jaw involved in activities such as chewing, speaking, and other functions. The TMJ is intricately connected to the autonomic nervous system, which governs various involuntary processes in the body. Autonomic fibers, comprising both sympathetic and parasympathetic fibers, extend their influence to the TMJ region. These fibers have the capacity to impact factors, such as blood flow, inflammation, and other physiological responses within the TMJ area. A previous case report showed improvement in motor dysfunction of PD after usage of bite splint together with oral medication [38]. Although the first scenario needs to be supported by new studies, the second scenario is a more widely accepted approach considering the symptoms of PD. The symptoms of PD including rigidity, muscular stiffness, and tremor of the masticatory muscles can lead to clenching, bruxism, and TMDs [18,19,39]. Moreover, considering that Parkinson’s disease primarily impacts the motor system, it is probable that the consequent motor symptoms directly contribute to the occurrence of TMDs.

Although it is impossible to imply whether TMJ asymmetry is a risk factor for PD or whether PD is a risk factor for TMJ asymmetry, the current study presents an association between PD and TMJ asymmetry. Based on the study’s results, preventive measures can be implemented clinically to address the risk factors for both TMDs and PD. Patients with PD can be made aware of the benefits of physical therapy for jaw exercises, making changes to their daily life habits and dietary habits, and managing bruxism. These interventions can contribute to improving patients’ overall well-being. Bruxism has been observed to alter activity in subcortical, cortical, and spinal circuits, as well as modify deep tendon reflexes in the extremities [40,41]. Previous studies have revealed that muscular tone in various regions of the body can be influenced by mechanosensory signals originating from the jaws, transmitted through the trigeminal nerve [40,42]. Therefore, clenching, bruxism, asymmetry of the TMJs, and morphological differences in the TMJs may be considered as potential risk factors associated with future PD. Consequently, regular examinations of the TMJs can be implemented as part of routine healthcare practices for healthy individuals.

Condylar position can vary among individuals with different skeletal patterns [43]. However, in the current study, the skeletal patterns of the subjects could not be considered due to the limited number of subjects with specific diseases who also met the selection criteria. Therefore, future studies could establish selection criteria based on skeletal patterns or subgroups classified according to different skeletal patterns. Hence, the measurements could be more sensitive by eliminating the influence of different condylar position in different skeletal patterns.

Previously acquired brain MR images were used in the current retrospective study. The primary purpose of obtaining these images was for neurological evaluation, not TMJ assessment. Therefore, the position of the mouth (open or closed) was not taken into consideration when acquiring these images. Although efforts were made to mitigate the potential impact of this factor by selecting the paired samples *t*-test and using of index values, in future prospective studies, it may be considered to acquire images with both an open and closed mouth to address this limitation.

The other limitations of present study were the confounding factors, which may also affect the morphology and symmetry of TMJ. These factors could not be considered during selection the subjects due to the lack of previous records and limited sample size of subjects with a specific disease. Factors such as dental history, TMD symptoms, current dental situation, craniofacial pattern, and malocclusion could be other factors that may affect the morphology and asymmetry of the TMJ, thereby influencing the results of the current study. Rather than using brain MR images, the MR images obtained specifically from the TMJ region would provide more accurate measurements. The records utilized in the present study were readily available for neurological assessments. However, when considering the relationship between PD and the TMJ, the inclusion of TMJ images has the potential to be incorporated into the standard initial records for Parkinson’s patients in the future.

To the best of our knowledge this was the first study that investigates the association between the TMJ and PD on a morphological basis. Current results could offer new study designs in order to assess the role of TMJ morphology in development of PD with a larger sample size and with less confounding factors.

## 5. Conclusions

The null hypotheses were partly rejected. Patients with PD showed asymmetry in their bilateral temporomandibular joints regarding to joint spaces in sagittal view, condyle head length, condyle neck width, the long axis of the condyle, the minor axis of the condyle, condylar inclination, and joint spaces in sagittal view. Healthy individuals demonstrated symmetry in their bilateral temporomandibular joints for all these parameters except for the long axis of the condyle. There was difference in temporomandibular joint symmetry between healthy individuals and those with PD in terms of joint spaces in sagittal view, condyle head length, condyle neck width, the minor axis of the condyle, condylar inclination, and joint spaces in sagittal view.

## Figures and Tables

**Figure 1 diagnostics-13-02482-f001:**
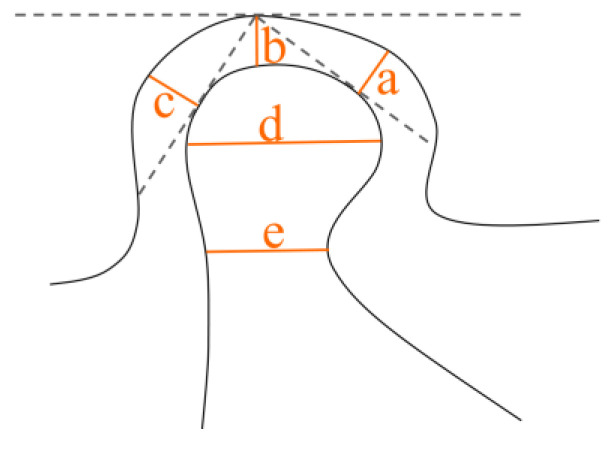
Parameters in sagittal view. a. AJS, b. SJS, c. PJS, d. CHL, e. CNW.

**Figure 2 diagnostics-13-02482-f002:**
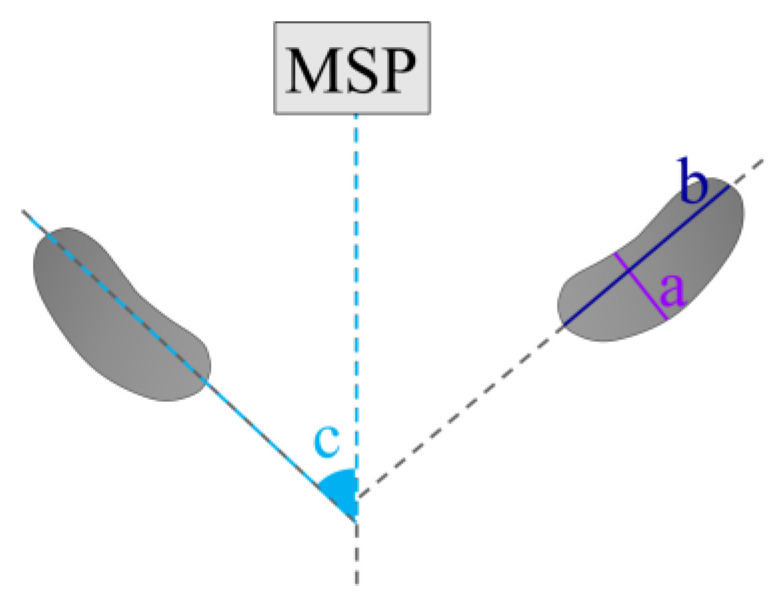
Parameters in axial view. a. MAC, b. LAC, c. CI.

**Figure 3 diagnostics-13-02482-f003:**
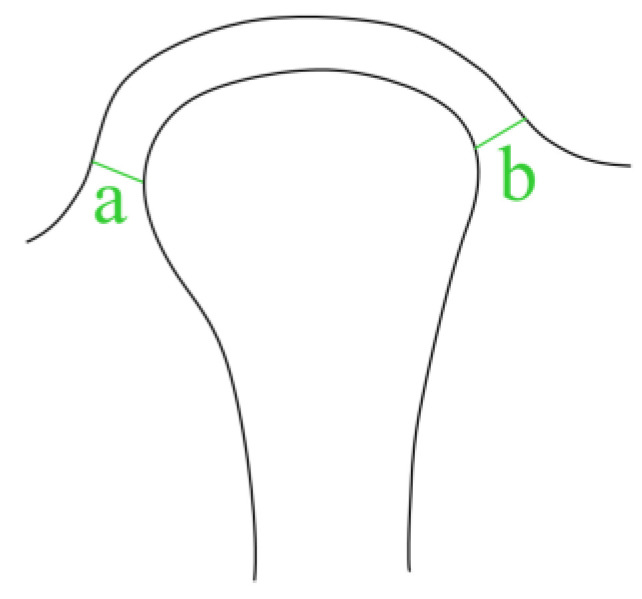
Parameters in coronal view. a. MJS, b. LJS.

**Figure 4 diagnostics-13-02482-f004:**
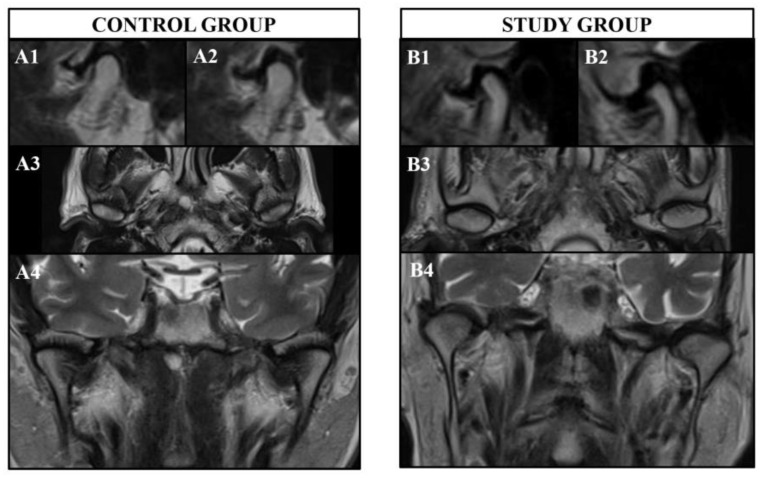
Images of bilateral TMJs from the study and control groups: (**A1**,**A2**) Healthy appearance of TMJ in sagittal view with symmetry of the right and left sides. (**A3**) Healthy appearance of TMJ in axial view with symmetry of the right and left sides. (**A4**) Healthy appearance of TMJ in coronal view with symmetry of the right and left sides. (**B1**,**B2**) Unhealthy appearance of TMJ in sagittal view with resorptive areas of the condyle head. Flattening of the condyle surface is observed in both TMJs, but it is more severe in (**B2**) compared to (**B1**) resulting in an asymmetrical appearance. An osteophyte is present at the anterior region of the condyle in (**B2**) further contributing to the asymmetry. (**B3**) Unhealthy appearance of TMJ in axial view with resorptive areas in each condylar head but in different localizations, resulting in an asymmetrical appearance of the right and left sides. (**B4**) Unhealthy appearance of TMJ in coronal view with asymmetry of right and left sides. An osteophyte is observed on the medial surface of the condyle on the right hand side. There is an increase in intensity of the inferior part of the lateral pterygoid muscle on left hand side, which may be related to myospasms.

**Table 1 diagnostics-13-02482-t001:** Description of the landmarks and parameters.

	Name	Abbreviation	Description
General Landmarks	Sella	Se	The central point of the pituitary fossa of sphenoid bone
	Nasion	Na	The contact point between frontal bone, right nasal bone, and left nasal bone
	Basion	Ba	The central point on the anterior margin of the foramen magnum
	Menton	Me	Most inferior midpoint of the chin on the outline of the mandibular symphysis
	Midsagittal Plane	MSP	A plane passes through the points Se, Na, and Ba
	Superior Glenoid Fossa	SGF	The point that is placed at the most superior wall of glenoid fossa
	Anterior Tangent Line	ATL	A line connects SGF and the most prominent anterior aspect of the condyle
	Posterior Tangent Line	PTL	A line connects SGF and the most prominent posterior aspect of the condyle
Sagittal View	Anterior Condyle Point	ACP	The most anterior point of the condyle, which is situated on the anterior tangent line
	Superior Condyle Point	SCP	The most superior point of the condyle
	Posterior Condyle Point	PCP	The most posterior point of the condyle, which is situated on the posterior tangent line
	Anterior Joint Space	AJS	The perpendicular distance from ACP to glenoid fossa
	Superior Joint Space	SJS	The perpendicular distance from SCP to SGF
	Posterior Joint Space	PJS	The perpendicular distance from PCP to glenoid fossa
	Condyle Anterior Point	CAP	Most anterior point of the condyle corresponding to the area of maximum condyle length
	Condyle Posterior Point	CPP	Most posterior point of the condyle corresponding to the area of maximum condyle length
	Condyle Head Length	CHL	The distance between CAP and CPP
	Anterior Neck Point	ANP	Deepest point on the anterior aspect of condylar neck
	Posterior Neck Point	PNP	Deepest point on the posterior aspect of condylar neck
	Condylar Neck Width	CNW	Distance between ANP and CNP
Axial View	Minor Axis of the Condyle	MAC	The maximum diameter of condylar process in anteroposterior direction
	Long Axis of the Condyle	LAC	The maximum diameter of condylar process in mediolateral direction
	Condylar Axis Inclination	CI	The angle between midsagittal plane and the long axis of the condyle
Coronal View	Medial Condyle Point	MCP	Most medial point of the condylar head
	Lateral Condyle Point	LCP	Most lateral point of the condylar head
	Medial Joint Space	MJS	The shortest distance between MCP and medial wall of the glenoid fossa
	Lateral Joint Space	LJS	The shortest distance between the most LCP and lateral wall of the glenoid fossa

**Table 2 diagnostics-13-02482-t002:** Inter-examiner reliability.

	Intraclass Correlation	95% CI		F Test		
		Lower Bound	Upper Bound	F	df	*p*
Right Side	0.907	0.861	0.920	25.691	49	0.001 *
Left Side	0.921	0.874	0.952	30.678	49	0.001 *

* *p*-value is less than 0.05.

**Table 3 diagnostics-13-02482-t003:** Demographic characteristics of the subjects.

		*N*	%	Age Range	Mean	S.D.
Study Group (PD)	Female	21	41.2%	52–90	73.10	8.60
	Male	30	58.8%	51–88	72.41	10.82
	Total	51	100%	51–90	72.70	9.85
Control Group	Female	20	39.2%	51–91	72.30	10.63
	Male	31	60.8%	53–87	72.68	9.69
	Total	51	100%	51–91	72.52	9.96

**Table 4 diagnostics-13-02482-t004:** Mean values of parameters for both TMJs in study and control groups.

	Study (*n* = 102)	Control (*n* = 102)
AJS	2.9 ± 1.3	3.4 ± 0.6
SJS	2.8 ± 1.3	3.5 ± 0.9
PJS	2.2 ± 1.0	2.3 ± 0.9
CHL	9.6 ± 1.8	10.0 ± 1.6
CNW	5.8 ± 1.4	6.0 ± 1.2
MAC	7.5 ± 1.6	8.0 ± 1.6
LAC	18.5 ± 2.5	18.8 ± 2.7
CI	68.67 ± 8.9	71.47 ± 7.14
MJS	2.7 ± 1.2	3.0 ± 0.7
LJS	2.6 ± 1.0	2.6 ± 0.7

**Table 5 diagnostics-13-02482-t005:** Comparison of bilateral jaws in study group through paired samples *t*-test.

		Side 1	Side 2	*p*
Sagittal	AJS	2.5 ± 1.0	3.4 ± 1.4	0.001 *
	SJS	2.4 ± 1.0	3.2 ± 1.4	0.001 *
	PJS	1.9 ± 0.8	2.5 ± 1.0	0.001 *
	CHL	8.8 ± 1.7	10.4 ± 1.5	0.001 *
	CNW	5.2 ± 1.2	6.5 ± 1.2	0.001 *
Axial	MAC	6.9 ± 1.5	8.1 ± 1.5	0.001 *
	LAC	17.8 ± 2.4	19.2 ± 2.3	0.001 *
	CI	64.72 ± 7.68	72.61 ± 8.32	0.001 *
Coronal	MJS	2.2 ± 1.0	3.3 ± 1.2	0.001 *
	LJS	2.1 ± 0.7	3.1 ± 0.9	0.001 *

* *p*-value is less than 0.05.

**Table 6 diagnostics-13-02482-t006:** Comparison of bilateral jaws in control group through paired samples *t*-test.

		Side 1	Side 2	*p*
Sagittal	AJS	3.2 ± 0.6	3.5 ± 0.6	0.100
	SJS	3.4 ± 0.9	3.6 ± 0.9	0.157
	PJS	2.3 ± 0.8	2.4 ± 0.9	0.243
	CHL	9.8 ± 1.6	10.2 ± 1.7	0.095
	CNW	5.9 ± 1.1	6.2 ± 1.1	0.210
Axial	MAC	7.8 ± 1.5	8.2 ± 1.7	0.190
	LAC	18.5 ± 2.9	19.0 ± 2.4	0.011 *
	CI	70.66 ± 6.96	72.29 ± 7.29	0.067
Coronal	MJS	2.9 ± 0.7	3.1 ± 0.6	0.254
	LJS	2.5 ± 0.7	2.7 ± 0.7	0.281

* *p*-value is less than 0.05.

**Table 7 diagnostics-13-02482-t007:** Comparison of asymmetry indices between study and control groups through Student’s *t*-test.

		Study (*n* = 51)	Control (*n* = 51)	*p*
Sagittal	AJS-Index	16.39 ± 10.62	3.76 ± 2.39	0.001 *
	SJS-Index	15.27 ± 12.15	3.97 ± 2.30	0.001 *
	PJS-Index	13.77 ± 10.30	3.67 ± 3.08	0.001 *
	CHL-Index	8.82 ± 7.86	1.99 ± 1.62	0.001 *
	CNW-Index	12.19 ± 11.44	2.52 ± 1.28	0.001 *
Axial	MAC-Index	8.36 ± 6.97	2.40 ± 5.97	0.001 *
	LAC-Index	3.91 ± 3.21	1.81 ± 6.68	0.064
	CI-Index	5.74 ± 3.87	1.13 ± 1.47	0.001 *
Coronal	MJS-Index	19.03 ± 14.28	3.95 ± 3.53	0.001 *
	LJS-Index	20.82 ± 14.94	4.65 ± 4.62	0.001 *

* *p*-value is less than 0.05.

## Data Availability

Research data are available upon request due to privacy or ethical restrictions.

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
