# Peer review of "Evaluation of Temporomandibular Joint in Patients with Parkinson’s Disease: A Comparative Study"

_diagnostics, 2023, doi:10.3390/diagnostics13152482_

Round 1

Reviewer 1 Report

Based on the provided manuscript, the study focuses on comparing TMJ morphology and asymmetry between healthy individuals and those with Parkinson's Disease.

The manuscript provides a clear introduction, methodology, and statistical analysis. However, there are a few suggestions for improvement:

- Provide a clear objective statement in the abstract: The abstract should include a concise statement of the study's objective. Currently, the objective is only implied. Please revise the abstract to explicitly state the objective of the study.

- Add details about participant characteristics: Include information about the age range, gender distribution, and any other relevant demographic characteristics of the participants in both the study and control groups.

- Improve clarity in the methodology: Provide more details about the measurement method, including the specific steps and landmarks used for each parameter. This will help readers understand how the measurements were performed.

- Clarify the statistical analysis: Provide additional information on the statistical tests used, including the assumptions made and any adjustments for multiple comparisons, if applicable.

- Discuss the clinical outcomes

- ADD THE REFEREnCES

Please, to reduce the risk of bias in Your reference list and avoid any risk of auto-citation read and use the fi-index tool.

The fi-index tool (Springer Nature Published) measures the impact of a paper on an author's career and it is obtained by a simple calculation that could be made with an online tool (www.fident.eu/fidentresearch/fiindextool). The use of the fi-index tool could be useful as a guarantee parameter on a specific manuscript provided that a particular author could have a scientific research trend. It is hoped that this index will be used on a large scale for scientific publications affected by bibliometric parameters. 

I took the liberty of carrying out the calculation for you, so it will be enough to add a subparagraph at the end of the results section with the relative citations, if the authors prefer it they can further cite this method also in the materials and methods section.

X.X: Fi-index tool

This manuscript has been checked with the Fi-index tool and obtained a score of 0.88 for the first author only on the date 20/02/2023 according to SCOPUS® [1,2]. The fi-index tool aims to ensure the quality of the reference list and limit any autocitations.”

1. Fi-Index: A New Method to Evaluate Authors Hirsch-Index Reliability. Publishing Research Quarterly 2022, 10.1007/s12109-022-09892-3, doi:10.1007/s12109-022-09892-3.

2. The Use of Fi-Index Tool to Assess Per-manuscript Self-citations. Publishing Research Quarterly 2022, 10.1007/s12109-022-09920-2, doi:10.1007/s12109-022-09920-2.

Minor editing of English language required

Author Response

Dear Reviewer,

Required changes were done in the manuscript and indicated within the text as ‘red’ letters.

The authors would like to sincerely thank for your valuable time and for the contributions to the study.

Reviewer 2 Report

The title of this study is TMJ Evaluation of Patients with Parkinson’s Disease: Three-Dimensional Comparative Study. However, the 10 parameters measured in this study were linear measurements on 2D data only. The TMJ morphology was evaluated using sagittal, axial, and coronal views which are all in 2D views. No 3D evaluation at all. Please change the title accordingly.

Page 2, line 71. This study aimed to perform a 3D evaluation of the TMJ in patients with PD… what kind of 3D evaluation?

Page 3, line 96, 98 and 99. The authors claimed that they did 3D morphological comparisons for the first, second, and third part of the study. What kind of 3D morphological comparisons?  Linear measurements on 2D data whether it’s in sagittal view (Figure 1), axial view (Figure 2), and coronal view (Figure 3), they are still in 2D views. Therefore, the authors cannot claim it as 3D evaluation unless they converted the data into 3D images and measured from there.

Page 4, line 166. In the third part of the study, an unpaired t-test was employed to compare the asymmetry indices of each TMJ parameter…. However, in Page 5, line 180, and Page 6, Table 6, the authors mentioned student t-test to compare the asymmetry indices between study and control groups. Please be consistent of the term used.

Page 7, Figure 4. Label the images and explain about the images in the caption rather than explaining in general.

No references found in this 9 pages manuscript.

Moderate editing needed

Author Response

(The authors gave the same response as above.)

Reviewer 3 Report

The paper is well-structured and provides a clear overview of the study. However, there are a few potential issues that could be addressed;

  1. The study design is retrospective, which could introduce biases and limit the interpretation of causal relationships.
  2. It is not clear what the clinical significance of the findings is, or how they could be used to inform treatment or management of PD.
  3. The inclusion and exclusion criteria could be more clearly defined. While the paper states that subjects with craniofacial anomalies, menton deviation greater than 2 mm, a history of traumatic injury, or a history of surgery in the craniofacial region were excluded, it is not clear how these criteria were assessed and measured.
  4. The statistical analysis could be more detailed. While the paper describes the statistical tests that were used, it does not provide information on the assumptions that were made for each test (e.g. normality of data), or on the adjustments that were made for multiple comparisons.
  5. The paper could provide more information on the selection of the control group. While it states that healthy subjects with MR images of the brain that included the TMJ area were selected, it does not provide information on how these subjects were recruited or screened.

6.     In the discussion, more thoroughly compare and contrast the results with previous studies. Mention both similarities and differences. Consider possible reasons for differences. Discuss the limitations of the study more critically and comprehensively. Other potential limitations beyond the ones mentioned could likely be identified.

  1. Limit the conclusions to what the results actually demonstrated. The current conclusion overstates what the study actually showed.
  2. Avoid causal claims if the study design does not allow determining causality. A retrospective study cannot establish if TMJ morphologycauses or contributes to PD.

Author Response

(The authors gave the same response as above.)

Round 2

Reviewer 2 Report

The authors have addressed all issues.

Minor editing of English language